# Genomic Characterization of *Arcobacter butzleri* Strains Isolated from Various Sources in Lithuania

**DOI:** 10.3390/microorganisms11061425

**Published:** 2023-05-28

**Authors:** Dainius Uljanovas, Greta Gölz, Susanne Fleischmann, Egle Kudirkiene, Neringa Kasetiene, Audrone Grineviciene, Egle Tamuleviciene, Jurgita Aksomaitiene, Thomas Alter, Mindaugas Malakauskas

**Affiliations:** 1Department of Food Safety and Quality, Faculty of Veterinary Medicine, Veterinary Academy, Lithuanian University of Health Sciences, Tilzes St. 18, LT-47181 Kaunas, Lithuania; dainius.uljanovas@lsmu.lt (D.U.);; 2Institute of Food Safety and Food Hygiene, Freie Universität Berlin, Königsweg 69, 14163 Berlin, Germany; 3Statens Serum Institut, Artillerivej 5, DK-2300 Copenhagen, Denmark; 4Kaunas Clinical Hospital Microbiology Laboratory, Medical Academy, Lithuanian University of Health Sciences, Josvainiu St. 2, LT-47144 Kaunas, Lithuania; 5Department of Pediatrics, Medical Academy, Lithuanian University of Health Sciences, Eiveniu St. 2, LT-50161 Kaunas, Lithuania

**Keywords:** *Arcobacter butzleri*, genomic diversity, pangenome, antimicrobial and heavy metal resistance, virulence genes, whole-genome sequencing

## Abstract

*Arcobacter (A.) butzleri*, the most widespread species within the genus *Arcobacter*, is considered as an emerging pathogen causing gastroenteritis in humans. Here, we performed a comparative genome-wide analysis of 40 *A. butzleri* strains from Lithuania to determine the genetic relationship, pangenome structure, putative virulence, and potential antimicrobial- and heavy-metal-resistance genes. Core genome single nucleotide polymorphism (cgSNP) analysis revealed low within-group variability (≤4 SNPs) between three milk strains (RCM42, RCM65, RCM80) and one human strain (H19). Regardless of the type of input (i.e., cgSNPs, accessory genome, virulome, resistome), these strains showed a recurrent phylogenetic and hierarchical grouping pattern. *A. butzleri* demonstrated a relatively large and highly variable accessory genome (comprising of 6284 genes with around 50% of them identified as singletons) that only partially correlated to the isolation source. Downstream analysis of the genomes resulted in the detection of 115 putative antimicrobial- and heavy-metal-resistance genes and 136 potential virulence factors that are associated with the induction of infection in host (e.g., *cadF*, *degP*, *iamA*), survival and environmental adaptation (e.g., flagellar genes, CheA-CheY chemotaxis system, urease cluster). This study provides additional knowledge for a better *A. butzleri*-related risk assessment and highlights the need for further genomic epidemiology studies in Lithuania and other countries.

## 1. Introduction

The *Arcobacter* species are Gram-negative, motile bacteria belonging to the *Arcobacteraceae* family (phylum *Campylobacterota*) [1]. Since the proposal of the *Arcobacter* genus in 1991, it has been subjected to reclassification, and its taxonomic organization is still under debate [2]. Recently, Pérez-Cataluña et al. [3] reassessed the taxonomy of *Arcobacter* genus using phylogenetic and genomic analyses and proposed to divide it into seven different genera. Although the novel genus names were validated, authors of a subsequent study [4] refuted the proposal by demonstrating that *Arcobacter* is a phenotypically, phylogenetically, and genomically coherent taxon. Currently, the genus *Arcobacter* is a large group accommodating 36 recognized species that can be detected in a plethora of habitats [5]. 

*Arcobacter* (*A*.) *butzleri* is the most widespread species of the genus *Arcobacter* and is considered an emerging zoonotic enteropathogen, with contaminated water, food of animal origin, and vegetables as the most likely sources of infection for humans [6]. *A*. *butzleri*-caused illness in humans has been associated with acute or prolonged diarrhea (lasting from >2 weeks to 2 months), abdominal pain, nausea, vomiting, and, in some cases, bacteremia, peritonitis, and endocarditis [6,7]. Although *Arcobacter*-caused intestinal and extra-intestinal infections appear to be self-limiting, the severity and persistence of the symptoms might require the use of antimicrobial agents [8]. However, *A. butzleri* isolates— from animals, food products, environment, and human clinical samples—frequently (20–100%) display a multidrug-resistant profile, hampering the treatment of infections [9,10]. A recent long-term survey taken in Belgium ranked *A. butzleri* as the fourth most common bacterial pathogen isolated from the fecal samples of patients with enteritis [11]. The pathomechanism in *Arcobacter* infection and putative virulence factors are not yet completely understood. A previous in vitro survey [12] showed that *A*. *butzleri* induces a loss of barrier function in the epithelial monolayers of HT-29/B6 cells by changes in tight-junction proteins (claudin-1, -5, -8) and induction of cell apoptosis. This leak–flux mechanism is consistent with the watery type of diarrhea in *A*. *butzleri* infections [12]. Other studies have demonstrated that *A*. *butzleri* possesses adhesive, invasive, and cytotoxic properties, and can induce the expression of the proinflammatory cytokine interleukin-8 (IL-8) in Caco-2 and IPI-2I cell lines [13,14].

The first complete *A. butzleri* genome sequence (from the human strain RM4018) was published in 2007 by Miller et al. [15]. After analyzing the genomic data of RM4018, the authors determined the presence of putative resistance and virulence determinants homologous to those found in other pathogens (e.g., *lrgAB*, *cat*, *cadF*, *cj1349*, *ciaB*, *pldA*, *tlyA*, *mviN*, *hecA*, *hecB*, *irgA*, *iroE*) [15]. Since then, a few other whole-genome sequencing (WGS)-based studies were conducted to elucidate the genetic mechanism behind *A. butzleri* resistance and virulence [8,10,16,17,18]. Recently, Müller et al. [19] presented two databases that include the nucleotide sequences of all known *A. butzleri* putative virulence and antimicrobial- and heavy-metal-resistance genes. Despite these advancements, data on *A. butzleri* genomic characteristics and its overall zoonotic potential in relation to antimicrobial susceptibility is scarce.

Therefore, with this study, we aimed to perform a comparative WGS analysis to determine the genetic diversity, pangenome structure, putative virulence, and antimicrobial- and heavy-metal-resistance genes of 40 Lithuanian *A. butzleri* strains that previously underwent antimicrobial-susceptibility testing.

## 2. Materials and Methods

### 2.1. A. butzleri Strains

A total of 40 *A. butzleri* strains (see Appendix A) were included in this study. All strains were collected and identified at the species level using molecular methods (multiplex polymerase chain reaction [PCR] and *rpoB* sequencing) during a previous *Arcobacter* prevalence study in Kaunas, Lithuania [20]. All 40 strains were also previously tested for susceptibility to six antimicrobial agents (ampicillin, ciprofloxacin, gentamicin, tetracycline, azithromycin, and erythromycin) using the gradient strip diffusion method [20]. Half of *A. butzleri* strains were isolated from different food products obtained from local retail markets: raw cow milk (*n* = 11), chicken meat (*n* = 7) and ready-to-eat (RTE) salad mixes (*n* = 2). The remaining strains originated from human stool (*n* = 10) and environmental water samples (*n* = 10, including lake and river-water samples). All 40 strains were stored at −80 °C in a brain–heart infusion broth (BHI) (Oxoid, Thermo Fisher Scientific, Basingstoke, UK) containing 30% (*v*/*v*) glycerol (Stanlab, Lublin, Poland). Before genomic DNA (gDNA) extraction, *A. butzleri* strains were cultured on Mueller–Hinton agar (Oxoid, Thermo Fisher Scientific) plates supplemented with 5% (*v*/*v*) defibrinated sheep blood (MHB) (Oxoid, Thermo Fisher Scientific) at 30 °C for 48 h under microaerobic conditions.

### 2.2. Genome Sequencing and Assembly

The gDNA was extracted with the MasterPure^TM^ Complete DNA and RNA Purification Kit (Lucigen Corporations, Middleton, WI, USA) according to the manufacturer‘s instructions. The resulting gDNA purity, integrity and concentration were assessed by NanoDrop^TM^ 2000 (Thermo Fisher Scientific, Wilmington, DE, USA), electrophoretic run on 1% agarose gel and Qubit 2.0 Fluorometer (Life Technologies, Thermo Fisher Scientific, Darmstadt, Germany) using the double-stranded DNA (dsDNA) assay HS kit (Invitrogen, Thermo Fisher Scientific, Waltham, MA, USA), respectively. After the assessment, gDNA was used to construct Nextera XT sequencing libraries (llumina, San Diego, CA, USA) according to the manufacturer’s instructions. Libraries were then subjected to paired-end sequencing (2 × 300 bp), which was performed using an Illumina MiSeq^TM^ system at the Institute of Microbiology and Epizootics (IMT), Department of Veterinary Medicine, Freie Universität Berlin, Berlin, Germany. Adapter trimming was done with Illumina Experiment Manager software (v.1.18.1). Reads were de novo assembled into contigs with SPAdes v.3.12.0 [21] setting the ‘*-careful*’ option to reduce mismatches and short indels. To improve the overall quality of the assemblies, contigs of less than 500 bp long were removed from the genomes using a filtering tool (v.1.1.2) within KBase platform [22]. Subsequently, quality-associated assembly statistics, such as the number of contigs, total genome bp length and the number of uncalled bases (Appendix A), were calculated using QUAST v.5.0.2 [23].

### 2.3. Bioinformatic Analysis

The assembled genomes (in FASTA format) were submitted to KmerFinder v.3.2 web tool [24] from the Center for Genomic Epidemiology (CGE) (Technical University of Denmark [DTU]; https://cge.food.dtu.dk/services/KmerFinder/; accessed on 20 February 2022) for a contamination check and species prediction. To assess the genetic divergence, the average nucleotide identity (ANI) between the genome sequences of Lithuanian strains and the genome sequences of reference strains *A. butzleri* RM4018 and *A. trophiarum* LMG 25534^T^ was calculated using the Python module pyani v.0.2.9 (ANIblastall algorithm) [25]. The genomes of *Campylobacter jejuni* subsp. *jejuni* NCTC 11168^T^ and *Helicobacter pylori* NCTC 11637^T^ were included as an outgroup. In addition, digital DNA–DNA hybridization (dDDH) was performed using the genome-to-genome distance calculator (GGDC) (ggdc.dsmz.de; accessed on 20 February 2022) [26] for species delimitation and to calculate differences in G+C genomic content. In this study, formula 2 (the sum of identical base pairs over all high-scoring segment pairs (HSPs) divided by the total length of all HSPs) was preferred for the calculation of DDH as it does not consider genome lengths and is thus recommended [26] for the analysis of draft genomes.

The phylogenetic relatedness among the *A. butzleri* strains included in this study was determined using CSI phylogeny pipeline v.1.4 [27] available on the CGE server (https://cge.food.dtu.dk/services/CSIPhylogeny/; accessed on 1 March 2022). During analysis, the paired-end reads were aligned to the reference genome of *A. butzleri* strain RM4018. Genomic positions, containing a single nucleotide polymorphism (SNP) in at least one of the strains, and meeting quality-filtering criteria in all strains, were included in the SNP matrix. The following criteria were used for SNP quality-filtering: (i) a minimum depth of 10 reads at SNP positions, (ii) a minimum of 10% of the average depth at SNP positions, (iii) a minimum distance of 10 bp between each SNP, (iv) a minimum SNP-quality score of 30, (v) a minimum read mapping quality of 25, and (vi) a minimum Z-score of 1.96. The SNPs that did not meet the criteria were excluded from further analysis. The retained genomic positions were concatenated per isolate, and their alignments were subjected to maximum likelihood tree construction using FastTree v.2.1.7 [28]. The obtained trees were visualized using either iTOL (https://itol.embl.de/; accessed on 12 September 2022) [29] or FigTree (http://tree.bio.ed.ac.uk/software/figtree/; accessed on 4 March 2022).

Gene prediction and annotation was performed within Galaxy platform [30] using the software tool Prokka v.1.14.6 with standard settings [31]. The resulting GFF3 files were used as input for the pangenome analysis with Roary v.3.13.0 [32]. The tool was run with default settings and the identity threshold to cluster protein homologues was set at 90%. The accessory binary tree alongside the binary gene presence and absence matrix were visualized using Phandango [33]. In addition, the *gene_presence_absence.csv* output file was used to find orthologous genes. The associations between binary presence/absence data of all genes in the accessory genome and the main sources of isolation (human stool, chicken meat, raw cow milk and environmental water) were assessed with Scoary v.1.6.16 [34] and were considered significant if *p*-value (Benjamini–Hochberg-corrected) < 0.05.

All genomes were screened for known antimicrobial resistance (AMR) genes against ResFinder, NCBI, CARD, ARG-ANNOT and MEGARes databases [35,36,37,38,39] using a BLASTn-based search with ABRicate v.1.0.1 (https://github.com/tseemann/abricate; accessed on 9 April 2022). The tool was used with the default settings, except for the minimum DNA %identity (‘*-minid*’) and minimum DNA % coverage (‘*-mincov*’), which were set to 75 and 50, respectively. In addition, ABRicate was utilized to screen all genomes against a custom database (ARCO_IBIZ_AMR; https://gitlab.com/FLI_Bioinfo_pub; accessed on 29 April 2022) [19], that was created specifically for *Arcobacter*, and which contains potential AMR genes (*n* = 92) and putative heavy-metal-resistance genes (*n* = 27). To detect mutations in the 23S rRNA, *rplD*, *rplV* and *gyrA* genes, these regions were extracted and aligned with Geneious Prime^®^ 2022.1.1 by global alignment using default parameters. The presence of plasmids was predicted using PlasmidFinder v.2.1 (https://cge.food.dtu.dk/services/PlasmidFinder/; accessed on 20 May 2022) [40]. Putative-virulence factors were identified with ABRicate software using a custom *Arcobacter* database (ARCO_IBIZ_VIRULENCE; https://gitlab.com/FLI_Bioinfo_pub; accessed on 29 April 2022) [19], which involves, besides others, genes (*n* = 148) associated with motility, chemotaxis, adhesion, invasion, hemolysis, iron absorption, lipid A biosynthesis, and type IV secretion system (T4SS). Virulence-, AMR-, and heavy-metal-resistance gene profiles were visualized on a heat map using Morpheus software (https://software.broadinstitute.org/morpheus/; accessed on 11 November 2022).

## 3. Results and Discussion

### 3.1. Assembly Results and Whole-Genome-Based Taxonomic Classification of Strains 

The genome features of 40 *A. butzleri* strains are summarized in Appendix A. De novo genome assemblies resulted in 18 to 244 contigs per strain. In accordance with previous studies [10,18], the obtained draft genomes displayed a mean GC content of 26.9% (ranging from 26.79 to 27.11%) and a mean length of 2.29 Mb (ranging between 2.09 and 2.46 Mb). PlasmidFinder v.2.1 was used to determine if the genome-length variation could be attributed to the presence or absence of plasmids. No plasmid replicons were identified by the tool in any of the 40 tested genomes. 

During a k-mer-based analysis with KmerFinder v.3.2, all assembled genomes did match to genomic sequences of different *A. butzleri* reference strains (i.e., RM4018, ED-1, 7h1h and JV22). Therefore, all isolates were taxonomically classified as *A. butzleri*. In order to confirm the predicted species, genome-wide ANI and dDDH analyses were performed. 

The results of the ANI analysis revealed that all 40 Lithuanian strains and the reference strain *A. butzleri* RM4018 together form a cluster (Figure 1). ANI values for every pair of the Lithuanian strains ranged from 96.62 to 99.99% (mean, 97.57%) (Figure 1, Appendix A). High pairwise similarity (>99%) was mainly observed for strains that were associated with the same isolation source. However, the closest relatives for human strain H19 were the raw-milk strains RCM42, RCM80 and RCM65 (99.99%, 99.98% and 99.96% ANI, respectively), and for H26, the strains RCM60 and RCM74 (99.51% and 99.48% ANI, respectively). Interestingly, the nearest relatives for *A. butzleri* reference strain RM4018 were water strains W48, W50 and W44 (98.17%, 98.13% and 98.04% ANI, respectively). Furthermore, the Lithuanian strains shared a mean ANI of 97.49% (ranging between 96.98 and 98.26%) with the reference genome RM4018. In contrast, all *A. butzleri* strains showed lower average nucleotide identity (77.18%, 65.86% and 63.13%) with *A. trophiarum* LMG 25534^T^, *C. jejuni* subsp. *jejuni* NCTC 11168^T^ and *Helicobacter pylori* NCTC 11637^T^, respectively (Appendix A).

DDH analysis confirmed the clustering obtained by ANI analysis, with a mean pairwise value of 79.49% (ranging from 76.20 to 84.10%) between the *A. butzleri* reference genome RM4018 and the Lithuanian strains (Appendix A). The comparison of the *A. butzleri* reference genome RM4018 with *A. trophiarum* LMG 25534^T^ and with outgroup genomes *C. jejuni* subsp. *jejuni* NCTC 11168^T^ and *Helicobacter pylori* NCTC 11637^T^ resulted in low dDDH values (20.90%, 21.80% and 25.60%, respectively). According to other authors [41,42], the recommended ANI and dDDH values for species delineation are 95~96 and 70%, respectively. Thus, the results of our study indicate that all Lithuanian strains are sufficiently related to be assigned to the *Arcobacter* gen. nov. as *A*. *butzleri* comb. nov. [43,44].

### 3.2. Phylogenetic Analysis of Lithuanian A. butzleri Strains

The core genome single nucleotide polymorphism (cgSNP) analysis using the strain RM4018 as reference was based on 1,543,017 (65.91%) nucleotide positions that were common to all genomes. As shown in Figure 2, the analysis resulted in the strain clustering into two major clusters (I and II). Both clusters were intermixed with strains of different origins. The smaller cluster (cluster I) consisted of 10 strains, which were isolated from chicken meat (*n* = 4), environmental water (*n* = 3), RTE salad mixes (*n* = 2) and human stool (*n* = 1). Meanwhile, cluster II consisted of 30 strains obtained from raw cow milk (*n* = 11), human stool (*n* = 9), environmental water (*n* = 7) and chicken meat (*n* = 3). Strains of cluster I differed from strains of cluster II by 10,296 to 11,735 SNPs (mean, 11,306). The SNP variation was similar within both clusters, with maximum differences of 9857 and 9751 SNPs between strains in clusters I and II, respectively. We were not able to link the clustering pattern of Lithuanian *A. butzleri* strains with isolation sources or dates. However, this was expected, as *A. butzleri* exhibits a relatively small (between 1165 and 1651 genes) but highly diverse core genome [10,18]. Recently, after performing a pangenome analysis of 49 *A. butzleri* strains, Isidro et al. [10] reported that at least 55% of the core loci presented ≥40 alleles.

Although the clustering pattern of *A*. *butzleri* was not related to the isolation source, we identified seven groups (Figure 2; groups 1–7) that were composed of closely related isolates (≤5 SNPs) of shared origin, indicating that these were isolates of the same strain rather than different strains [45]. Group 1 belong to cluster I and consisted of two isolates (S41 and S42; pairwise difference of 3 SNPs) from different salad samples of the same variety. Similarly, a previous multi-locus sequence typing (MLST)-based analysis of *A*. *butzleri* from RTE vegetables demonstrated that several isolates, derived from different samples, shared the same sequence type [46]. However, one explanation for this low genetic diversity could be that all their samples were collected from the same processing plant [46]. Meanwhile, the remaining six groups belong to cluster II. Group 2 consisted of two water isolates (W19 and W43) that differed by 2 SNPs only. Both isolates derived from lake-water samples that were taken at the same public-bathing site (PBS2) two months apart. Therefore, it can be hypothesized that *A*. *butzleri* can persist in aquatic environments. The remaining eight water strains that were isolated from sampling sites PBS1 (*n* = 3) and PBS3 (*n* = 5) showed higher genetic divergence (9496–11,389 and 2159–11,526 SNPs, respectively). In addition, the grouping of these strains was not related to the sampling source and site. The comparative analysis of these results is limited, as there are no other whole-genome sequencing (WGS)-based studies that were performed to determine the genetic relatedness of environmental water strains. However, a previous analysis of a set of *A*. *butzleri* isolates from water using comparative genomic fingerprinting resulted in isolate clustering that was not related to sampling site [47]. According to other authors [48], the lower similarity of strains that share the same sampling site might indicate multiple sources of contamination. In contrast to our water strains, the grouping of strains isolated from raw milk was mostly associated with sample origin i.e., group 3 consisted of isolates derived from Farm C (RCM62 and RCM70; 4 SNPs), while the isolates of group 4 derived from Farm A (RCM39 and RCM45; 1 SNP) and the isolates of group 6 from Farm B (RCM60 and RCM74; 1 SNP). In addition, three other milk isolates (RCM42, RCM65 and RCM80) related to Farm A belong to a separate group (group 7). A maximum difference of five SNPs was observed between the milk isolates of group 7, while the minimum distance between group 4 and group 7 was 8,298 SNPs. Similarly, a previous pulsed-field gel electrophoresis (PFGE)-based study showed that different *A*. *butzleri* pulsotypes can be found in the same bulk-tank milk sample or different bulk-tank milk samples that were collected in the same farm during separate samplings [49]. The lower genetic similarity between the strains of group 4 and group 7 can be explained by multiple sources of milk contamination on the farm or by the presence of multiple *A*. *butzleri* strains in a single dairy animal, as suggested by Giacometti et al. [50]. This is also probably the reason why RCM63 (Farm A) and RCM69 (Farm B) did not group with the rest of the strains from a shared origin. It is worth mentioning that milk strains within groups 3, 4, 6 and 7 were isolated from samples that were collected during different months, indicating the ability of *A*. *butzleri* to adapt to a farm environment and persistently contaminate milk. Interestingly, group 7 also includes the human strain H19, which exhibits a maximum of four SNPs to the milk isolates within this group. Furthermore, the cgSNP comparison of H19 and RCM80 showed no differences. Although we do not have the medical record to directly link H19 with RCM42, RCM65 and RCM80, the shared grouping of these strains leads us to speculate that the human isolate most likely originated from milk. Therefore, this result supports the previous considerations [6,49] that raw cow milk is a potential source of human *A*. *butzleri* infections. Finally, group 5 consisted of two isolates (H14 and H18) from human stool that showed no pairwise SNP differences. Considering the fact that, during the present study, a proportion of strains of shared origin (RTE salads, milk, water) were genetically closely related, this finding might indicate that H14 and H18 originated from the same infection source.

### 3.3. Pangenome Analysis of A. butzleri Strains from Various Sources in Lithuania

In order to identify genes that were shared and distinct among the 40 *A. butzleri* strains, a pangenome analysis with Roary v.3.13.0 was performed (Figure 3). The analysis showed that the pangenome of the Lithuanian *A. butzleri* strains consisted of 7986 protein-coding genes. Of these, 18.52% (*n* = 1479) were classified as core genes (genes present in 99% ≤ genomes ≤ 100%) and 2.79% (*n* = 223) as soft-core genes (genes present in 95% ≤ genomes < 99%). The core genome covered approximately 58.34% of the average *A. butzleri* genome size (2.29 Mb). Meanwhile, the accessory genome represented 78.69% of the pangenome and was comprised of shell genes (*n* = 956; genes present in 15% ≤ genomes < 95) and cloud genes (*n* = 5328; genes present in <15% of genomes). Out of 6284 accessory genes, 47.80% (*n* = 3004) were classified as singletons (i.e., strain-specific genes). Similarly, during two recent studies [10,18], the core and accessory genomes comprised from 15.59 to 24.93% and from 75.07 to 84.41% of the *A*. *butzleri* pangenome (containing from 6623 to 7474 loci), respectively. It is also noteworthy that during our analysis, the total number of genes increased with the addition of new genomes (Figure 3A). Meanwhile, the core-genome size decreased reaching a plateau at approximately 10 genomes (Figure 3B). These results indicate that *A*. *butzleri* harbors an open pangenome, potentially resulting from a sympatric lifestyle with recurrent events of horizontal gene transfer (HGT) [18,51].

In contrast to the cgSNP phylogeny (Figure 2), the hierarchic tree based on the accessory genome revealed isolate clustering into four major clusters (Figure 3C; clusters I–IV). However, both trees did not show a clear segregation between strains that were isolated from different sources. Furthermore, the accessory binary tree revealed isolate grouping into the same seven groups (Figure 3C), as revealed by cgSNPs. According to the gene presence/absence matrix, isolates belonging to group 7 had 181 unique coding sequences (CDS) (total length of 169 kb), which were not present in the remaining *A. butzleri* strains (Figure 3C). The 181 CDS were mostly (77.90%; 141/181) annotated as hypothetical proteins by Prokka. According to other authors [10,18], *A*. *butzleri* possess a highly diverse core genome and an accessory genome that can include multiple polymorphic genes. In order to determine if the above-mentioned region contains polymorphic genes, we subjected the list of the 181 group 7-associated genes to local BLASTn analysis against the remaining 36 *A*. *butzleri* genomes. The analysis revealed that a total of 37 (20.44%) genes were present (coverage ≥ 55%; identity ≥ 74.64%) in strains that were not included in group 7. A previous retrospective genomic analysis of 197 *Salmonella enterica* serovar Dublin isolates from cattle showed that a clade-specific DNA region can indicate the presence of a plasmid [52]. However, during the current survey, group 7-specific genes (individually or in clusters) were found to be located on multiple contigs (regardless of analyzed strain), which only partially matched the unique DNA region (coverage ranging from 0.13 to 9.63%). Moreover, as mentioned above, we were not able to detect a plasmid replicon using PlasmidFinder. Hence, it is unlikely that the unique CDS observed for group 7 isolates were associated with the presence of a plasmid sequence. Nonetheless, group 7-specific CDS contained other mobile genetic elements, namely the prophage integrase IntA, the putative ATP-binding protein IS5376 and a transposase of the IS1634 family. Furthermore, after screening the contigs of strain RCM80, we noticed that a few clusters of group 7-specific genes were located next to different transposases (ISBmu3, ISSde5, ISMtsp22, ISVsa19 and IS1302). Based on these results, we concluded that the group 7-associated CDS partially resulted from horizontal gene transfer and gene polymorphism. However, the origin of the remaining unique genes and their effects on strain phenotype remain unclear.

### 3.4. Whole-Genome-Based Detection of Putative AMR- and Heavy-Metal-Resistance Genes

Out of the 40 *A. butzleri* strains that were used in this study, 24 (60%) showed phenotypic resistance to at least one antimicrobial agent of two or more classes (Table 1). However, the screening of *A. butzleri* genomes against public AMR databases (CARD, ARG-ANNOT, MEGARes, ResFinder and NCBI) resulted only in the prediction of three putative OXA-type β-lactamase gene groups (*bla_OXA-464-like_*, *bla_OXA-490-like_*, *bla_OXA-491-like_*) that are associated with resistance to a single class of antimicrobials (i.e., aminopenicillins). Similarly, Müller et al. [19] did not detect any AMR genes after screening the genomes of two *A. butzleri* strains that were phenotypically resistant to multiple antimicrobial agents. The authors have hypothesized that the prediction of genotypic resistance in *Arcobacter* species might be limited due to lack of known antimicrobial resistance-associated genes [19]. Therefore, further analysis was based on a custom *A. butzleri* database (ARCO_IBIZ_AMR; https://gitlab.com/FLI_Bioinfo_pub; accessed on 29 April 2022) that enabled the detection of putative AMR- and heavy-metal-resistance genes.

As depicted in Figure 4, the hierarchic tree based on the resistance gene presence/absence profiles showed clustering into the same seven groupsm as revealed by the cgSNP and pangenome analyses. After screening 40 *A. butzleri* genomes against ARCO_IBIZ_AMR database, we were able to detect 115 out of 119 (96.64%) genes. Around one third of the identified genes (45/115; 39.13%) encoded structural components of 18 putative efflux pump (EP) systems. Eleven of these transporters (EP2-EP6, EP8, EP10, EP12-15) were detected in all *A. butzleri* genomes. Similarly, Isidro et al. [10] detected ten EP systems (EP2, EP4–EP6, EP8, EP10, EP12–EP15) in all studied *A*. *butzleri* strains (*n* = 49). 

According to other authors [54], the three mechanisms of β-lactam resistance in Gram-negative bacteria are: (i) the production of β-lactamases; (ii) the production of novel penicillin-binding proteins (PBPs) with reduced affinity to β-lactam antibiotics; and (iii) the regulation of membrane permeability. During our survey, genomes were screened for the presence of four β-lactamase encoding genes (*bla1*-*bla3*, *hcpC*) and three genes (*mrdA*, *pbpB*, *pbpF*) that encoded PBPs. Surprisingly, five of these genes, namely *bla2*, *hcpC*, *mrdA*, *pbpB* and *pbpF*, were detected in all genomes (including the three genomes of strains that were susceptible to ampicillin). Similarly, Müller et al. [19] reported the presence of *bla2*, *hcpC*, *mrdA*, *pbpB* and *pbpF* genes in both tested *A. butzleri* strains that were susceptible to ampicillin but showed resistance to cefotaxime (β-lactam antibiotic in the third-generation class of cephalosporins). As result, the authors speculated that these genes are more associated with resistance to cefotaxime rather than ampicillin. In accordance with other authors [10], we observed an association between a reduced susceptibility to ampicillin and the presence of class D β-lactamase, as 94.59% (35/37) of the resistant strains harbored the *bla3* gene. The multialignment of class D β-lactamase protein (*n* = 35) showed that 62.86%, 31.43% and 5.71% of sequences share a mean amino acid identity of >99% with *bla_OXA-464_*, *bla_OXA-491_* and *bla_OXA-490_*, respectively. In contrast, the *bla1* gene was only detected in two genomes of water strains (5%) that were resistant to ampicillin. Thus, *bla1* might be associated with resistance to another β-lactam antibiotic that was not tested in this study.

Fluoroquinolone resistance in Gram-negative bacteria is mostly caused by target-specific point mutations within the quinolone-resistance-determining region (QRDR) of the *gyrA* gene [55]. In *A. butzleri*, the resistance to ciprofloxacin is associated with Thr-85-Ile and Asp-89-Tyr (equivalent to Thr-86-Ile and Asp-90-Tyr in *Campylobacter gyrA*) substitutions [10,56]. The multialignment of 40 *gyrA* gene sequences revealed that only the ciprofloxacin-resistant strain CH64 carried a point mutation (C254T) in the QRDR, which resulted in amino-acid substitution at position 85 (Thr-85-Ile). Therefore, our result supported the hypothesis that ciprofloxacin resistance could be mediated by a target-specific point mutation (Thr-85-Ile) in the *gyrA* gene.

In general, tetracycline resistance can be caused due to: (i) the activity of tetracycline-specific efflux pumps; (ii) mutations in the binding sites of 16S rRNA; or (iii) the activity of ribosomal protection proteins (RPPs). In *Campylobacter* spp., resistance to tetracycline is conferred by the *tet(O)* gene, which encodes a RPP [57]. For *A. butzleri*, the data on mechanisms causing tetracycline resistance are still limited. A study by Fanelli et al. [16] showed that one out of two tetracycline-resistant *A. butzleri* strains carried a gene encoding the tetracycline efflux protein TetA. Meanwhile, during another study in Germany, the *tetA* and *tet(O)* genes were not detected among tetracycline-resistant and susceptible *A. butzleri* strains from Muscovy ducks [19]. Within our survey, *tetA* was detected in 60% (24/40) of genomes, while *tet(O)* was not present in any of them. Although *tetA* was identified in a majority of genomes, there was no correlation with phenotype, as the gene was carried by both tetracycline-resistant (12/24; 50%) and susceptible strains (12/24; 50%). Furthermore, we screened our genomes for the presence of a previously reported elongation factor (ABU_RS09920) that shares the same C-terminus domain with the RPPs [16]. However, the elongation-factor-encoding sequences were identified in all 40 genomes. Hence, *A. butzleri* resistance to tetracycline might be mediated by a different mechanism. 

Macrolide (i.e., erythromycin and azithromycin) resistance is mainly due to target-specific mutations within the peptidyl-encoding region in domain V of the 23S rRNA gene and/or amino-acid substitutions in the ribosomal proteins L4 and L22 [58]. None of the 40 *A. butzleri* strains harbored a point mutation at positions 2085-6 in the 23S rRNA gene (corresponding to positions 2074-5 in *C. jejuni*), although 22 (55%) strains were resistant to at least one of the macrolides. Furthermore, none of the 40 Lithuanian strains exhibited described alterations in proteins L4 or L22 [58,59]. The presence of a putative macrolide efflux system (EP3), enrolling the two macrolide-export proteins MacA1 and MacB2, was reported in *A. butzleri* by two previous studies [8,19]. Regardless of macrolide-resistance phenotype, we were able to identify EP3 in all strains. It was hypothesized by other authors [8,19] that EP3 can be detected in macrolide-susceptible strains but might not be expressed or could have amino-acid substitutions that may cause lack of functionality. Furthermore, Isidro et al. [10] reported a correlation between reduced erythromycin susceptibility and truncation of the EP16 regulator (TetR). In our study, a full-length protein (179 amino acids (aa)) was detected in majority (21/22; 95.45%) of macrolide resistant strains, while a truncated TetR protein (122-aa) was determined in one strain (W46). The truncation resulted from a nonsense mutation at position 369 (G369A). It is worth mentioning that in comparison to the remaining strains with macrolide resistance, W46 showed the highest minimum inhibitory concentration (MIC) values for both azithromycin (64 µg/mL) and erythromycin (48 µg/mL). Therefore, it might be that the TetR size changes are associated with a higher level of resistance to erythromycin (MIC ≥ 32 µg/mL) and azithromycin (MIC ≥ 64 µg/mL).

There is growing concern that exposure of bacterial populations to heavy metals can facilitate the dissemination of antimicrobial resistance via co-selection of AMR genes and metal-resistance genes [16]. In our survey, we were able to detect all (*n* = 27) putative heavy-metal-resistance genes included in the ARCO_IBIZ_AMR database, although the *cnrA* gene (encoding the nickel and cobalt resistance protein CnrA) was exclusively present in one strain from water (W48). All Lithuanian *A. butzleri* strains carried the previously reported arsenic cluster, which is composed of the *arsABC* operon [16,19]. The additional screening of genomic sequences resulted in the detection of a gene coding for a transcriptional regulator ArsR (annotated as a hypothetical protein by Prokka) located next to ArsB in all strains. ArsR is a DNA-binding transcriptional repressor, which regulates its own expression and that of the remaining genes in the *ars* operon [60]. 

Furthermore, all strains harbored the ABC-type molybdate transporter (composed of the *modABC* operon) and its regulator *modE* [61]. This result is in agreement with previous studies [8,16]. Of note, the coding region of the ATPase ModC was initially annotated as a vitamin B12 import ATP-binding protein BtuD by Prokka. We also identified six additional transport proteins in all strains: (i) cadmium, zinc and cobalt transporting ATPase CadA, (ii) magnesium and cobalt efflux protein CorC, (iii) manganese-exporting P-type ATPase CtpC, (iv) cadmium, cobalt and zinc/H(+)-K(+) antiporter CzcD, (v) mercuric transporter MerT, and (vi) zinc transporter ZntB. This finding is in line with previous studies [16,19].

Moreover, all investigated strains encoded for five genes potentially involved in copper resistance, namely *copA1* (encoding the copper-exporting P-type ATPase A), *copA2* (encoding the putative copper-importing P-type ATPase A), *copZ* (encoding the copper chaperone CopZ), *csoR* (encoding the copper-sensing transcriptional repressor CsoR) and *cusS* (encoding a sensory kinase), which is in agreement with Müller et al. [19]. These authors also detected the *copR* gene (encoding the transcriptional activator protein CopR) in both tested *A. butzleri* genomes. Meanwhile, we detected *copR* in only 25 (62.50%) of our strains. The difference in *copR* detection rates can be explained by the lower number of genomes analyzed in the previous study. Indeed, after screening the genomes of five *A. butzleri* strains that were previously determined as copper-resistant [16], we were able to determine a similar *copR* prevalence rate (60%). This result also indicates that CopR might not be required for copper resistance in *A. butzleri*.

According to other authors [15,16], a homolog of the CzcCBA efflux pump is encoded by *A. butzleri*, although CzcC is absent in the genome. During our survey, we detected *czcA* and *czcB* genes encoding for cobalt–zinc–cadmium-resistance proteins in 7.50% (3/40) and 32.50% (13/40) of strains, respectively. Furthermore, we were able to identify five genes (*czcR1*, *czcR2*, *czcR3*, *czcR4*, *czcR5*) encoding transcriptional activators CzcR. The genes *czcR1*, *czcR2* and *czcR3* were detected in all strains, while *czcR4* and *czcR5* were present in 12.50% and 15% of the strains, respectively. It is worth mentioning that none of the strains carried all five CzcR-encoding genes simultaneously. Similarly, other authors [19] detected the *czcR1*, *czcR2* and *czcR3* genes in both *A. butzleri* genomes that were included in the previous study, whereas they could not detect the genes *czcA*, *czcB*, *czcR4* and *czcR5*.

In brief, a wide variety of efflux-pump-related genes and other AMR- heavy-metal-resistance determinants were detected in the genomes of the Lithuanian *A. butzleri* strains. Additionally, we were able to link the ampicillin- and ciprofloxacin-resistance phenotypes with the presence of three class D β-lactamases (OXA-464-like, OXA-490-like and OXA-491-like) and Thr-85-Ile substitution in GyrA, respectively. However, the mechanisms underlying resistance to macrolides and tetracycline remain unclear and require further phenotypic-susceptibility testing combined with WGS-based analysis.

### 3.5. Whole-Genome-Based Detection of Putative Virulence Genes

As summarized in Figure 5, the screening of 40 Lithuanian *A. butzleri* genomes against 148 putative-virulence genes in ARCO_IBIZ_VIRULENCE database resulted in the detection of 136 genes (91.89%). The average number of genes per genome was 79 (ranging from 72 to 86), with the exception of strain W41, which carried a total of 120 virulence-associated genes. 

Flagellar motility is an important colonization factor for bacterial pathogens, enabling migration to, and movement within, mucus to reach microenvironments that are favorable for growth. Furthermore, flagellin, the subunit of the flagellar filament, is a primary target for the innate immune system [62]. Out of 136 putative-virulence genes identified in this study, 36 (26.47%) are required for the assembly of flagellum and are involved in motor function. These 36 flagellar genes were detected in all Lithuanian *A. butzleri* strains, which is in accordance with previous reports [10,18].

In general, bacterial motility is regulated by a chemotactic signaling system, which enables their movement towards environments that contain higher concentrations of beneficial, or lower concentrations of toxic, substances. After screening the genomic sequences of 49 *A. butzleri* strains from various sources, Isidro et al. [10] reported a 100% detection rate of eight putative chemotaxis-system genes (*cheA*, *cheB*, *cheR*, *cheV*, *cheW*, *cheY1*, *cheY2* and *cheY3*), which is in line with our study. In addition, during the present study, two homologs of *Campylobacter* spp. chemotaxis-associated genes *luxS* (encoding the S-ribosylhomocysteine lyase) and *ccp* (encoding cytochrome c551 peroxidase) were detected. The *luxS* gene was present in all genomes, while *ccp* was detected in 12.50% (5/40) of genomes. It is noteworthy that only *A. butzleri* strains from environmental water were carrying the *ccp* gene. A higher *luxS* detection rate (100%) in comparison to *ccp* (12.2%) was also reported in a previous whole-genome sequencing-based study [10].

During the present survey, all strains carried eight genes (*lpxA*, *lpxB*, *lpxC*, *lpxD*, *lpxH*, *lpxK*, *lpxP*, and *waaA*) potentially associated with lipid A biosynthesis. Lipid A (endotoxin), the hydrophobic anchor of lipopolysaccharide (LPS) or lipooligosaccharide (LOS), acts as a potent stimulator of the host innate immune system [63]. Furthermore, all *A*. *butzleri* genomes contained *rfaC* and *rfaF* genes encoding the heptosyltransferases I and II, respectively. According to other authors [16], orthologs of *rfaC* and *rfaF* are involved in the assembly and phosphorylation of the inner-core region.

In addition to the above-mentioned virulence factors that are associated with chemotaxis, flagellar and LPS/LOS synthesis, all Lithuanian *A*. *butzleri* strains carried: (i) several genes potentially involved in host cell adhesion, invasion and hemolysis, namely *cadF* (annotated as *oprF*) and *cj1349* (encoding for the outer membrane proteins CadF and Cj1349, which promote the binding of bacteria to intestinal epithelial cells), *degP* (encoding an adhesion facilitating periplasmic protein with chaperone activity), *ciaB* (encoding the *Campylobacter jejuni* invasion antigen B), *iamA* (encoding an invasion-associated protein), *tlyA* (encoding hemolysin) and *pldA* (encoding an outer-membrane phospholipase associated with lysis of erythrocytes); (ii) the *mviN* gene (annotated as *murJ*; encoding an inner-membrane protein essential for peptidoglycan biosynthesis); (iii) the *phoP3* gene (encoding the transcriptional regulatory protein PhoP3); (iv) two genes potentially associated with the iron-uptake system, namely *cirA2* (encoding colicin I receptor) and *fur* (encoding ferric-uptake regulator); (v) a VOC (vicinal oxygen chelate) family-virulence protein, (vi) the *cvfB* gene (encoding the conserved virulence factor B, which contributes to the expression of virulence factors and to pathogenicity in *Staphylococcus aureus*); and (vii) the *virF* gene (encoding the virulence-regulon-transcriptional activator VirF) [15,16,64,65,66,67]. Although the virulence factors *cadF*, *cj1349*, *ciaB*, *tlyA*, *pldA* and *mviN* were previously tested for their involvement in *A. butzleri* pathogenicity, the role of these and other core virulome genes (e.g., *degP*, *iamA*, *fur*, *cvfB*) in human infection remains unclear and requires further investigation [10].

Different detection rates were observed for the remaining virulence factors that are potentially involved in pathogenicity, adaptation, and immune evasion. During the present survey, we were able to detect two-component signal-transduction-system genes *phoP* (*phoP1*, *phoP2*, *phoP3*) encoding the transcriptional regulatory protein PhoP) and *phoQ* (encoding the sensor protein PhoQ). The PhoPQ system is used by Gram-negative bacteria to regulate the protein and lipid contents of the cell envelope. Lipid A modifications are of particular importance, as these structural alterations cause the bacteria to be less recognized and stimulatory to the TLR4 complex, and contribute to cationic antimicrobial peptide resistance [64,68]. Even though all *A. butzleri* strains carried the *phoP3* gene, *phoQ* was only detected in 14 (35%) strains. Furthermore, only 15% (6/40) of the strains carried all four genes (*phoP1*, *phoP2*, *phoP3*, *phoQ*) simultaneously. The putative-virulence genes *iroE* (alternatively annotated as *besA*) and *irgA* (annotated as *cirA1*) were detected in 28 (70%) genomes and mostly adjacent to each other. Genes *irgA* and *iroE* encode functional components (iron-regulated outer-membrane protein, and siderophore esterase) of the iron-uptake system in uropathogenic *E*. *coli* and therefore are required for establishing and maintaining infections [15]. Of note, we detected two variants of the *irgA* gene that varied in amino acid sequence and size. The first variant (671-aa) was present in majority (21/28; 75%) of strains, while the remaining seven strains (CH11, W20, W23, W33, W44, W46, W50) carried the second larger variant (696-aa). A recent genomic analysis [18] of 32 *A*. *butzleri* strains revealed similar *iroE* and *irgA* detection rates (75% and 81%, respectively). Besides *irgA* (*cirA1*) and *cirA2*, the ARCO_IBIZ_VIRULENCE database includes two additional *cirA* genes (*cirA3* and *cirA4*). Similar to *cirA1*, *cirA2* and *cirA3*, the product of *cirA4* was also annotated as colicin I receptor by Prokka, and therefore might be involved in iron acquisition [19]. *cirA3*, a *Campylobacter* spp. *cfrB* homolog, encoding the ferric enterobactin receptor CfrB that is involved in iron uptake [65], was present in eight (20%) genomes. Meanwhile, the *cirA4* gene was detected in five milk strains and one strain that was isolated from human stool. Interestingly, out of six strains that carried *cirA4*, four (H19, RCM42, RCM65 and RCM80) belong to the phylogenetic group 7. It is also worth noting that all five milk strains carrying the *cirA4* gene were isolated from milk produced by the same farm (Farm A). According to others [10,18], genes *hecA* (encoding an adhesin of the filamentous hemagglutinin family) and *hecB* (encoding hemolysin-activation protein) were found simultaneously and adjacent to each other in the *A*. *butzleri* genome. Although we detected both genes, the initial screening of contigs using ABRicate showed a considerably lower *hecA*-detection rate (2.50%; 1/40) in comparison to *hecB* (62.50%; 25/40). This discrepancy can be explained by the high polymorphism of *hecA*, which also causes false-negative amplifications during PCR-based screening [10]. Therefore, the nucleotide sequences of the *hecB* gene and flanking regions were compared with homologs in the NCBI database using the BLASTn algorithm. As a result, *hecA* (annotated as hypothetical protein) was detected in a total of 25 genomes next to *hecB* (annotated as *shlB*). In addition, during the present survey, we detected the *A*. *butzleri* urease cluster *ureD(AB)CEFG*, which consists of two structural genes (*ureAB* and *ureC*) and four accessory genes (*ureD*, *ureE*, *ureF*, *ureG*; encoding proteins that deliver the nickel to the urease and a nickel-uptake system), in 26 (65%) genomes. The presence of an urease cluster in *A*. *butzleri* was investigated in two other studies, which reported slightly different detection rates (ranging from 51.02 to 84.38%) [10,18]. There are no previous reports of *ureD(AB)CEFG* misannotation or polymorphism; therefore, differences in detection rates could be attributed to other factors, such as the amounts of genomes analyzed and the origins of isolates (i.e., genetic variation between strains isolated from different geographic locations and sources). It is noteworthy that during the present survey, all human strains carried the *ureD(AB)CEFG* cluster, while it was only present in a single strain from water. Out of the two above-mentioned comparative genomic studies, only the survey of Isidro et al. [10] included two strains from river water. Therefore, further studies are needed to analyze the association of urease clusters in *Arcobacter* isolates with their biological origins. According to other authors [10,15], the presence of urease cluster *ureD(AB)CEFG* and structural changes of encoded proteins (UreAB and UreC) are linked to the urease phenotype in *A*. *butzleri*. Hence, it is likely that several *A*. *butzleri* strains could utilize urease to produce ammonia through urea hydrolysis in order to survive in a low-pH environment [19]. Recently, Isidro et al. [10] identified a T4SS, which resembles a VirB/D4 secretion system, in one out of 49 *A. butzleri* strains. Likewise, we were able to detect this secretion system in a single strain (W41). The T4SSs are key virulence mediators in different human pathogens (e.g., *H. pylori*, *Brucella* spp., *Legionella pneumophila* and *Bordetella pertussis*), since they are associated with the modulation of host-cell apoptosis, cytotoxicity, intracellular survival, and manipulation of the host immune response [18].

Among the putative-virulence factors that were not detected in any of the Lithuanian *A*. *butzleri* strains are seven genes (*fcl*, *glmM2*, *gmhA2*, *gmhB2*, *hddA*, *hddC*, *rmd*) potentially linked with the synthesis of capsular polysaccharide. This result is in concordance with previous studies [10,19].

All these findings (i.e., the detection of a large repertoire of virulence genes associated with the induction of infection in host, survival and environmental adaptation) support the role of *A. butzleri* as a zoonotic pathogen. However, further comparative genomic analysis, as well as studies of virulence-gene expression in vitro and in vivo, are needed to unveil the mechanism of *A. butzleri* pathogenicity.

## 4. Conclusions

In summary, during the present study, we used a WGS approach to construct our own in-country database and provide first insights into the genetic diversity, mechanisms behind antimicrobial resistance, and virulence profiles of *A. butzleri* strains from various sources in Lithuania. The cgSNP analysis indicated a high genetic similarity between four *A. butzleri* strains that were isolated from raw cow milk (RCM42, RCM65, RCM80) and human stool (H19). These strains also showed a recurrent grouping pattern in all inferred hierarchical trees, which were based on accessory genome content, and virulence- and resistance-gene profiles. To the best of our knowledge, this is the first genomic survey to support previous considerations that raw milk is a potential *A. butzleri* transmission source. The pangenome analysis revealed a large and highly variable accessory genome that only partially correlated to the isolation source. In addition, the screening of genomic sequences revealed a wide variety of putative-virulence genes (associated with the induction of infection in host, survival and environmental adaptation) and potential AMR-heavy-metal-resistance determinants. Phenotypic *A. butzleri* resistance to ampicillin correlated with the presence of three class D β-lactamases (OXA-464-like, OXA-490-like, and OXA-491-like), while resistance to ciprofloxacin was linked to the presence of the Thr-85-Ile substitution in GyrA. Although our findings provided additional knowledge that may contribute to a better *A. butzleri*-related risk assessment, further genomic studies involving human and non-human isolates are needed to: (i) improve the usefulness of WGS analysis of the species; (ii) determine the genetic diversity within and/or between different geographic regions; and (iii) identify the emergence of novel virulence- and antimicrobial-resistance markers.

## Figures and Tables

**Figure 1 microorganisms-11-01425-f001:**
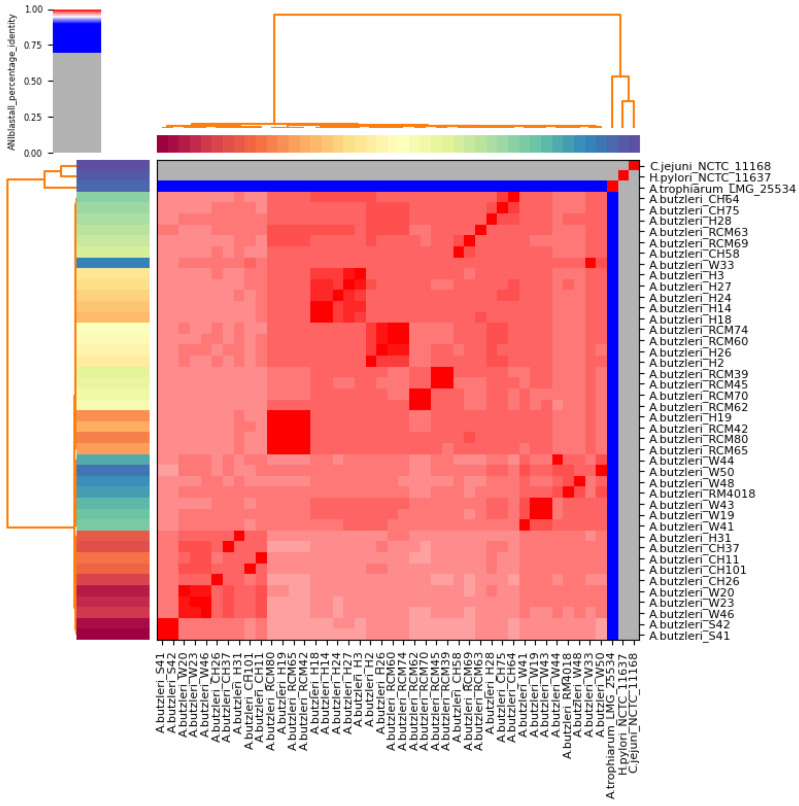
Heatmap visualization of the average nucleotide identity (ANI) between 40 Lithuanian *A. butzleri* strains, the *A. butzleri* reference strain RM4018, and the type strains of *A. trophiarum* (LMG 25534), *C. jejuni* subsp. *jejuni* (NCTC 11168), and *H. pylori* (NCTC 11637). Genomes with ANI > 95% (red cells) were considered to belong to the same species. For specific pairwise ANI values, refer to Appendix A.

**Figure 2 microorganisms-11-01425-f002:**
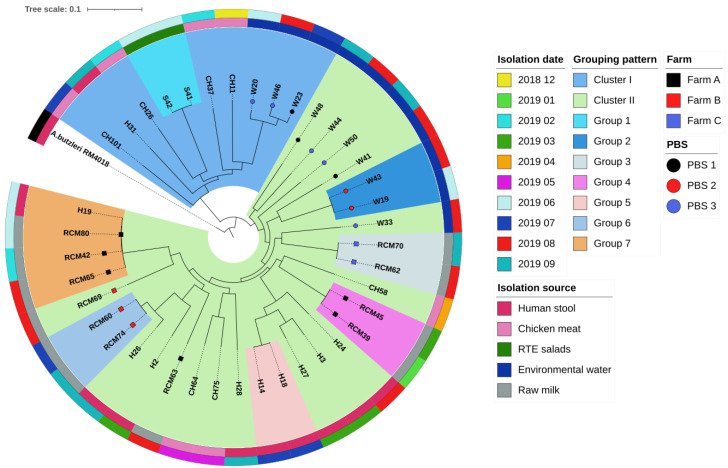
Phylogenetic tree of 40 Lithuanian *A. butzleri* strains based on whole genome single nucleotide polymorphism (SNP) analysis. The colour code of the outer circle represents the isolation date, of the middle circle the isolation source and of the inner circle the affiliation to clusters and groups. All isolates originating from milk produced in Farm A, B and C are labeled with black, red, and blue squares, respectively. The black, red, and blue circles at the tips of the branches indicate isolates from water samples collected at three different public bathing sites (PBS 1–3).

**Figure 3 microorganisms-11-01425-f003:**
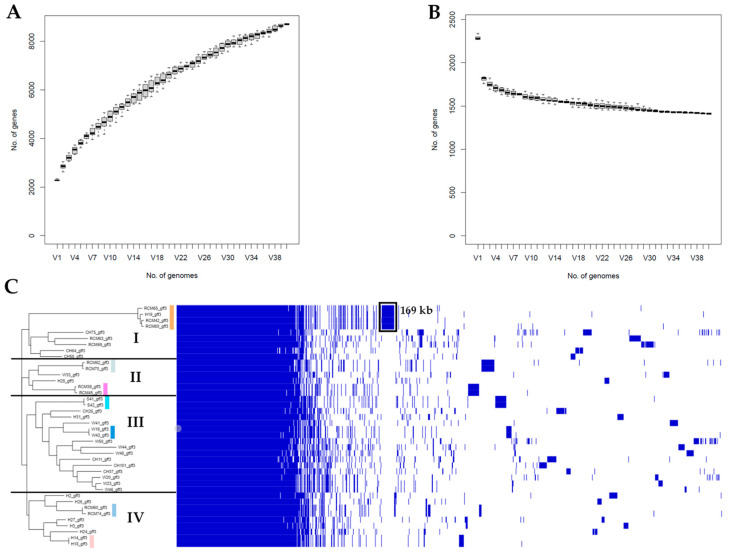
Pangenome of 40 *A. butzleri* strains isolated from various sources in Lithuania: (**A**) the graph shows the number of genes in the pangenome as increasing number of genomes are added in random order; (**B**) the number of conserved genes in the pangenome as genomes are added in random order; and (**C**) the hierarchic tree (left side) was generated based on the gene presence/absence matrix (right side). The clusters are denoted by numbers I to IV. Colored stripes indicate the seven groups as determined by core genome SNP (cgSNP) phylogeny (Figure 2). Each row in the matrix corresponds to a branch in the tree, while each column represents an orthologous gene family. Blue and white squares indicate gene presence and absence, respectively. The black square represents 181 coding sequences (CDS) that comprise the group 7-specific DNA region.

**Figure 4 microorganisms-11-01425-f004:**
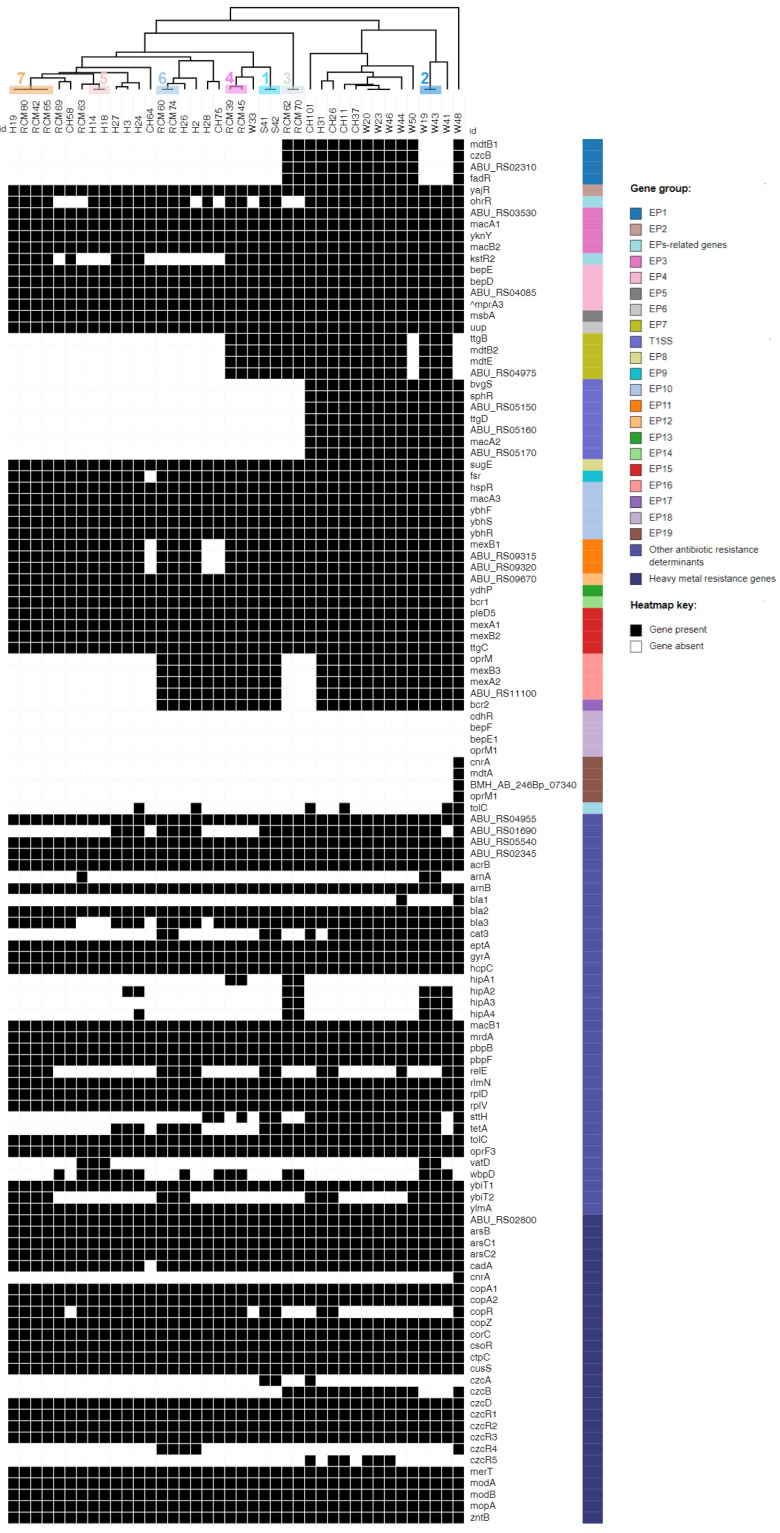
Heatmap showing the distribution of putative antimicrobial resistance and heavy-metal-resistance genes in each *A. butzleri* genome. Gene names or their locus tags are shown for each gene considered. ^An alternative transcriptional regulator (annotated as SlyA) of the RND (resistance-nodulation-division) system BepDE (EP4) was detected in eight strains isolated from environmental water (W20, W23, W44, W46, W50) and chicken meat (CH11, CH37, CH58). Numbers 1 to 7 (Pearson’s correlation-based hierarchical tree) indicate the seven groups that were observed in the cgSNP phylogeny and in the accessory genome-based hierarchic tree (Figure 2 and Figure 3, respectively). EP—efflux pump, T1SS—type I secretion system.

**Figure 5 microorganisms-11-01425-f005:**
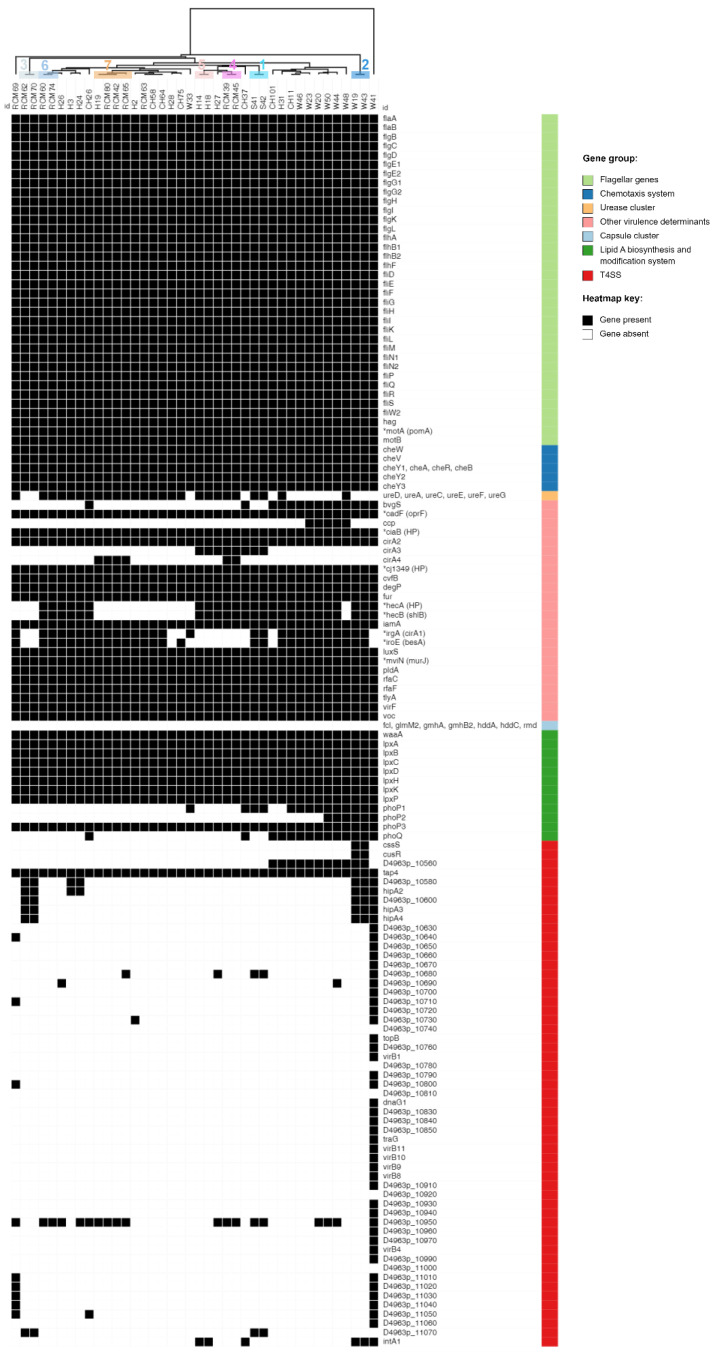
Heatmap showing the presence/absence of putative-virulence genes in each *A. butzleri* genome. Gene names or their locus tags are presented for each gene under consideration. *Genes which annotation was confirmed by BLASTn comparison against the NCBI database. The initial annotation is presented in brackets. Numbers 1 to 7 (dendrogram) indicate the seven groups of *A. butzleri* as defined by the cgSNP phylogeny and by the hierarchical trees that were based on accessory genome and resistance gene profiles. T4SS—type IV secretion system.

**Table 1 microorganisms-11-01425-t001:** Antimicrobial susceptibility of *A. butzleri* strains (*n* = 40).

Strain	MIC (µg/mL) (R/S)					
	AMP	AZM	GEN	TET	ERY	CIP
CH11	64 (R)	32 (R)	0.75 (S)	3 (R)	8 (S)	0.25 (S)
CH26	64 (R)	32 (R)	1.5 (S)	3 (R)	12 (R)	0.125 (S)
CH37	64 (R)	32 (R)	0.75 (S)	3 (R)	6 (S)	0.25 (S)
CH58	32 (R)	3 (S)	1 (S)	3 (R)	6 (S)	0.25 (S)
CH64	4 (S)	2 (S)	0.75 (S)	2 (S)	4 (S)	12 (R)
CH75	16 (R)	32 (R)	1.5 (S)	2 (S)	24 (R)	0.125 (S)
CH101	24 (R)	2 (S)	0.5 (S)	2 (S)	3 (S)	0.19 (S)
H2	24 (R)	1 (S)	0.5 (S)	0.75 (S)	1.5 (S)	0.047 (S)
H3	12 (R)	1.5 (S)	0.38 (S)	1 (S)	3 (S)	0.032 (S)
H14	8 (S)	2 (S)	1 (S)	2 (S)	4 (S)	0.125 (S)
H18	6 (S)	2 (S)	0.75 (S)	1.5 (S)	4 (S)	0.19 (S)
H19	12 (R)	2 (S)	1 (S)	1.5 (S)	4 (S)	0.125 (S)
H24	16 (R)	2 (S)	0.5 (S)	1.5 (S)	4 (S)	0.064 (S)
H26	12 (R)	12 (R)	1 (S)	1 (S)	6 (S)	0.064 (S)
H27	96 (R)	2 (S)	2 (S)	3 (R)	4 (S)	0.125 (S)
H28	12 (R)	24 (R)	0.75 (S)	3 (R)	8 (S)	0.19 (S)
H31	64 (R)	32 (R)	0.75 (S)	3 (R)	12 (R)	0.125 (S)
RCM39	12 (R)	32 (R)	2 (S)	2 (S)	12 (R)	0.25 (S)
RCM42	64 (R)	1.5 (S)	1 (S)	1.5 (S)	4 (S)	0.064 (S)
RCM45	12 (R)	24 (R)	1.5 (S)	2 (S)	12 (R)	0.19 (S)
RCM60	16 (R)	4 (S)	2 (S)	1.5 (S)	2 (S)	0.125 (S)
RCM62	32 (R)	1.5 (S)	1.5 (S)	1.5 (S)	0.75 (S)	0.064 (S)
RCM63	48 (R)	2 (S)	0.75 (S)	1.5 (S)	4 (S)	0.064 (S)
RCM65	64 (R)	2 (S)	0.75 (S)	1.5 (S)	4 (S)	0.064 (S)
RCM69	32 (R)	2 (S)	1.5 (S)	2 (S)	3 (S)	0.094 (S)
RCM70	24 (R)	2 (S)	2 (S)	1.5 (S)	2 (S)	0.125 (S)
RCM74	16 (R)	12 (R)	2 (S)	1.5 (S)	4 (S)	0.125 (S)
RCM80	48 (R)	2 (S)	1 (S)	1.5 (S)	6 (S)	0.19 (S)
S41	64 (R)	24 (R)	0.75 (S)	3 (R)	6 (S)	0.19 (S)
S42	64 (R)	32 (R)	0.75 (S)	3 (R)	12 (R)	0.19 (S)
W19	96 (R)	32 (R)	0.75 (S)	1.5 (S)	12 (R)	0.064 (S)
W20	48 (R)	32 (R)	1 (S)	3 (R)	6 (S)	0.19 (S)
W23	48 (R)	24 (R)	1 (S)	4 (R)	6 (S)	0.25 (S)
W33	96 (R)	32 (R)	1 (S)	2 (S)	12 (R)	0.125 (S)
W41	96 (R)	48 (R)	1 (S)	2 (S)	12 (R)	0.125 (S)
W43	192 (R)	16 (R)	0.75 (S)	2 (S)	8 (S)	0.064 (S)
W44	64 (R)	12 (R)	0.75 (S)	3 (R)	4 (S)	0.125 (S)
W46	48 (R)	64 (R)	0.5 (S)	4 (R)	48 (R)	0.75 (R)
W48	256 (R)	16 (R)	0.75 (S)	1.5 (S)	6 (S)	0.064 (S)
W50	96 (R)	24 (R)	1 (S)	3 (R)	12 (R)	0.19 (S)

Since no breakpoint values are available for *A. butzleri*, strains were classified as susceptible (S) or resistant (R) by comparing minimum inhibitory concentration (MIC) data with European Committee on Antimicrobial Susceptibility Testing (EUCAST) breakpoints for *Enterobacterales* (ampicillin and gentamicin) or *Campylobacter coli* (azithromycin, tetracycline, erythromycin, and ciprofloxacin) [53]. Resistance breakpoint (µg/mL): ampicillin (AMP) > 8; azithromycin (AZM) > 8; gentamicin (GEN) > 2; tetracycline (TET) > 2; erythromycin (ERY) > 8; ciprofloxacin (CIP) > 0.5.

## Data Availability

The assembled genomes were deposited to the NCBI database under the BioProject number PRJNA913680.

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
