# Peer review of "Genomic Characterization of Arcobacter butzleri Strains Isolated from Various Sources in Lithuania"

_microorganisms, 2023, doi:10.3390/microorganisms11061425_

Round 1

Reviewer 1 Report

This paper by Uljanovas and co-workers, entitled "Genomic characterization of Arcobacter butzleri strains isolated from various sources in Lithuania", reproduces the methodology described in previous works (i.e. Isidro J et al., 2020; Müller E et al., 2020) for the analysis of 40 isolated strains from species  A. butzleri.

While I find this work very interesting, since species from Arcobacter, as potential emerging pathogens, should be studied in-depth, I also wonder why authors did merely copied previous works for a new collection of isolates. I think that current genomic data on Arcobacter butzleri could have encouraged the authors for a more comprehensive comparative work. 

Minor comments:

Line 37: Epsilonproteobacteria have disapeared in the new 16S-based and genome-based taxonomies. Family Arcobacteraceae is now included in phylum Campylobacterota.

Table S1 (and results): Given that not a single genome was assembled in a single contig. How are the authors sure that the degree of genome completeness is suitable por a core / flexible genome analysis?

Line 91-92: What are "environmental water samples"? Could the authors be more precise? (freshwaters, rivers, wastewaters, etc.).

Line 126: While the dDDH analysis is described, I do not find information about ANI calculations.

Line 150: 90% minimum percentage identity for BLASTP analysis of, what?

Line 153: "genes that are shared...". How do the authors define an ortholog?

Line 182: Which is the genome length variation for the species? Why differences in genome length are only attributed to the presence/absence of plasmids? What happens with the flexible genome and strain-specific genes?

Line 219: It seems authors define "core genome" as the nucleotide positions which are common (identical?) in all genomes. Is it right? If so, how do authors consider othologs which are not 100% identical at the nucleotide level? How does microdiversity affect this collection of isolates?

Figure 2: Do the authors think that the SNP-based tree does really represent the phylogeny of the isolates? What if the flexible genome is included in the information? Is it more important a single SNP that the acquisition of a gene by horizontal gene transfer?

About the pangenome: which functions do the authors find in the flexible genome? Do authors think that cluster I (in figure 3A) should be separated? (the top branch semms deeply divergent).

Do the authors think that the genomic homogeneity of this collection of strains is due to culture conditions? It would be very interesting to see what metagenomes indicate about these populations.

Points 3.4 and 3.5 should be summarized.

Reviewer 2 Report

The manuscript focus on a comparative genome-wide analysis of 40 Arcobacter butzleri strains from Lithuania to determine the genetic relationship, pangenome structure, putative virulence and potential antimicrobial/heavy metal resistance genes. The following issues need to be improved before publication.

1.     When the abbreviation of a noun term appears for the first time in the text, its full name should be clearly stated such as AMR.

2.     Write clearly the criteria or threshold for judging the susceptible (S) or resistant (R) of the strains.

3.     The image clarity of Figure 3 should be adjusted.

4.     The results of metal resistance genes are compared with other studies, but there is a lack of analysis of the reasons.

5.     The writing details need to be improved such as the tenses of line 178 and 189, as well as reference 41.

6.     Sort illustrations according to the order in which the chart are mentioned in the paper. (Figure 3)

7.     The number of references in the past five years is less than 1/3 and should be updated.

Reviewer 3 Report

This article characterizes a set of 40 Arcobacter butzleri genomes.  It would be great to compare these 40 genomes with the other genomes currently available in GenBank.

 Unfortunately this is not so easy!

I would recommend the authors add a short section in the introduction about the name changes for this organism. Actually, this species has recently been moved to Aliarcobacter butzleri.  There have been LOTS of name changes recently, and this makes keeping track of things difficult (I think something like 80% of bacterial species have had their names changed in the past couple of years, according to genome taxonomy database!  ugh.) 

A good reference to mention for A. butzleri might be the following:

Daniele Chieffi, Francesca Fanelli, and Vincenzina Fusco, "Arcobacter butzleri: Up-to-date taxonomy, ecology, and pathogenicity of an emerging pathogen", COMPREHENSIVE REVIEWS IN FOOD SCIENCE AND FOOD SAFETY, 2020;1–39.   https://doi.org/10.1111/1541-4337.12577

I think that this is worth a couple of sentences in the introduction on this – because this problem is likely to be common for many microbiologists!  Unfortunately, this name change makes it difficult to figure out how many genomes there are currently in GenBank.  If I go to the 'old' NCBI table [https://www.ncbi.nlm.nih.gov/genome/browse/#!/prokaryotes/Arcobacter%20butzleri], there are 134 A. butzleri genomes available. 

If I go to the 'new' page, I get ZERO Arcobacter butzleri genomes, and 136 Aliarcobacter butzleri genomes [https://www.ncbi.nlm.nih.gov/data-hub/genome/?taxon=28197] - but one has to be aware of the name change - this is absolutely crazy that the NCBI pages don't have a synonym table for recognizing the old names!

I think that there should definitely be a reference to the name change, and reference a publication (such as the Daniele Cheffi paper mentioned above) that gives the new name, and perhaps a small discussion at the end about the difficulties in keeping track of bacterial species names!

I also think that, at a bare minimum, the authors should add a comparision with the known type strain for A. butzleri.  [see for example: https://www.namesforlife.com/10.1601/nm.3819 - I will paste information from the table below]

Name: doi 10.1601/nm.3819

Name: Arcobacter butzleri (Kiehlbauch et al. 1991) Vandamme et al. 1992

Name status:  Validly Published   current authority (Kiehlbauch et al. 1991) Vandamme et al. 1992

Preferred name: Aliarcobacter butzleri

Taxonomic rank: species

Nomenclatural type:   (designated strain)

Type strain  representative organism: D2686T (=NCTC 12481 =ATCC 49616 =LMG 10828 =RM4018 =CDC D2686)

Now when I go to the NCBI genome page:

https://www.ncbi.nlm.nih.gov/data-hub/genome/?taxon=28197

I can search in the strain names field, and find three genomes for the type strain:

1. Aliarcobacter butzleri NCTC 12481

https://www.ncbi.nlm.nih.gov/data-hub/genome/GCF_900187115.1/

chromosome LT906455

2. Aliarcobacter butzleri RM4018

https://www.ncbi.nlm.nih.gov/data-hub/genome/GCF_000014025.1/

chromosome CP000361

3. Aliarcobacter butzleri LMG 10828

https://www.ncbi.nlm.nih.gov/data-hub/genome/GCF_024584145.1/

LOCUS       JANJGJ010000000           27 rc    DNA     linear   BCT 07-AUG-2022

So I’d recommend including these three replicate genomes of the (same) type strain for A. butzleri, for a control comparison with the other 40 genomes.  This will allow for two tests / discussions – how does the type strains sequenced for the paper compare to the genomes from the known type strain for A. butzleri?  And as a kind of internal control, since two of these are complete as one contiguous piece, they SHOULD be pretty close to each other in the analysis.

Finally, a quick look at the quality of the genomes on the NCBI genome pages shows that there are only eight A. butzleri genomes (out of the 135 genomes) that are completely finished (as one contiguous piece), and one of these is marked as a ‘reference’ strain (although I’m not sure this is a TYPE STRAIN ?).  So I’d recommend including these genomes as well, although two of those are already included in the above type strain list, so this is only an additional six genomes.  I think that inclusion of these genomes would allow a nice comparison of the new genomes sequenced from Lithuania with those in GenBank.
